# Mutations That Confer Drug-Resistance, Oncogenicity and Intrinsic Activity on the ERK MAP Kinases—Current State of the Art

**DOI:** 10.3390/cells9010129

**Published:** 2020-01-06

**Authors:** Karina Smorodinsky-Atias, Nadine Soudah, David Engelberg

**Affiliations:** 1Department of Biological Chemistry, The Institute of Life Science, The Hebrew University of Jerusalem, Jerusalem 91904, Israel; karinasun@gmail.com (K.S.-A.); nadine.soudah@mail.huji.ac.il (N.S.); 2CREATE-NUS-HUJ, Molecular Mechanisms Underlying Inflammatory Diseases (MMID), National University of Singapore, 1 CREATE WAY, Innovation Wing, Singapore 138602, Singapore; 3Department of Microbiology, Yong Loo Lin School of Medicine, National University of Singapore, Singapore 117456, Singapore

**Keywords:** MAPK kinase, ERK1, ERK2, CD domain, Rolled, SCH772984, VRT-11E, *sevenmaker*

## Abstract

Unique characteristics distinguish extracellular signal-regulated kinases (Erks) from other eukaryotic protein kinases (ePKs). Unlike most ePKs, Erks do not autoactivate and they manifest no basal activity; they become catalysts only when dually phosphorylated on neighboring Thr and Tyr residues and they possess unique structural motifs. Erks function as the sole targets of the receptor tyrosine kinases (RTKs)-Ras-Raf-MEK signaling cascade, which controls numerous physiological processes and is mutated in most cancers. Erks are therefore the executers of the pathway’s biology and pathology. As oncogenic mutations have not been identified in Erks themselves, combined with the tight regulation of their activity, Erks have been considered immune against mutations that would render them intrinsically active. Nevertheless, several such mutations have been generated on the basis of structure-function analysis, understanding of ePK evolution and, mostly, via genetic screens in lower eukaryotes. One of the mutations conferred oncogenic properties on Erk1. The number of interesting mutations in Erks has dramatically increased following the development of Erk-specific pharmacological inhibitors and identification of mutations that cause resistance to these compounds. Several mutations have been recently identified in cancer patients. Here we summarize the mutations identified in Erks so far, describe their properties and discuss their possible mechanism of action.

## 1. Introduction

The unusual biochemical properties of the extracellular signal-regulated Kinases (Erks), their numerous biological functions and their critical roles in essentially all types of cancer, make these enzymes important subjects for research, and attractive targets for therapeutic purposes. Indeed, more than 50,000 studies have addressed aspects of the biochemistry, biology and pathology of Erks. Nevertheless, serious obstacles, which seem to be related to the unusual characteristics of the Erk enzymes, have been hindering the research. One of the hurdles has been the lack of key reagents, such as intrinsically/constitutively active mutants of Erks, and another is the absence of specific pharmacological inhibitors, not to mention clinically relevant inhibitors. The unavailability of these tools was unexpected, because useful inhibitors and a variety of active mutants were readily developed for most other protein kinases, including those that function upstream and downstream of Erk and those that are similar to Erks, such as p38s and JNKs [1,2,3,4,5,6,7,8,9,10,11,12,13,14,15,16].

This situation has been changing dramatically in the last decade. An arsenal of over a dozen useful inhibitors was finally developed, and, soon after, numerous mutations that render Erks resistant to these drugs were identified. Other mutations in ERKs were found in screens for cells in culture that acquired resistance to inhibitors of Raf and MEK. The mutations that cause drug resistance joined a small number of mutations that had been generated on the basis of gain-of-function mutations in lower organisms, or via structural studies. Finally, sequencing of genomes of tens of thousands of cancer patients led to the discovery of a few more mutations in ERKs.

Thus, a large number of interesting mutations in ERKs has been finally gathered (Table 1). The effects of many of these mutations on the structure, biochemistry, biology, or pathology of Erks have not yet been fully characterized, but some notions are emerging. This review summarizes our current knowledge of ERK mutations and describes their effect on the catalytic, physiological, pharmacological and pathological properties of Erks.

### 1.1. The Erk MAP Kinases

#### 1.1.1. The Erk MAP Kinases Are Conserved in All Eukaryotes and Carry Out a Plethora of Functions

Erk proteins form a small subgroup within the family of MAP kinases. In mammals this group is encoded by two genes, ERK1 and ERK2, and by several splicing variants thereof. Erk1 and Erk2 are expressed in all cells of the organism and are critical for the functionality of all tissues and body systems. An indication of the remarkable competency of Erks is the large number of substrates they phosphorylate—497 have been identified so far [17]. For comprehensive reviews on the Erks, see [18].

Erks are highly conserved in evolution structurally and functionally, so that many discoveries with Erks’ orthologs of *S. cerevisiae* (Fus3, Kss1 and Slt2/Mpk1; [19,20,21,22]) or of *D. melanogaster* (Rolled; [23,24]) are directly relevant to the mammalian molecules.

Mammalian Erk1 and Erk2 share 83% sequence identity and 88% similarity (alignment of the human proteins) and seem to be equally activated in response to relevant signals, suggesting that many of their activities are redundant. Observations that raised the possibility of distinct functions for each isoform were made primarily with knockout mice [25,26,27,28,29,30,31,32,33,34]. The most significant finding in this regard was that knocking out *ERK2* resulted in embryonic lethality, whereas knocking out *ERK1* had only mild effects [35]. Yet, overexpression of Erk1 in mice knocked-out for *ERK2*, restores viability and the mice are normal and fertile [36]. It seems, therefore, that the physiological functions of Erk1 and Erk2 are almost fully redundant and the dramatic difference in the phenotype between ERK1^−/−^ and ERK2^−/−^ mice stems solely from the fact that in most tissues Erk2 is expressed at much higher levels than Erk1 [36]. Also pointing to similarity in structure-function relationships are the observations that the majority (but not all) of the mutations identified recently and discussed in this review confer similar effects on the Erk1 and Erk2 proteins. Finally, the newly developed pharmacological inhibitors manifest similar (but not identical) efficacy towards the two isoforms, although it should be noted that some of those inhibitors have not yet been tested against both isoforms. These observations combined suggest minor differences in functionality between Erk1 and Erk2 native proteins.

#### 1.1.2. Erks Are Targets of the Proto-Oncogenic RTK-Ras-Raf-MEK Pathway

Erks function as the downstream targets of the receptor tyrosine kinase (RTK)-Ras-Raf-MEK pathway, which regulates a large number of biological processes in all cell types and in all developmental stages (for reviews on the RTK-Ras-Raf-MEK pathways see [37,38,39,40,41]). Although in particular cell-types and under some conditions Raf and MEK may phosphorylate various substrates [17], in response to most signals Erks seem to be the only targets of this cascade and therefore mediate most, if not all, of the effects of the pathway [42]. Erks are activated by all 20 subfamilies of RTKs [43], including the clinically important epidermal growth factor receptors (EGFRs), nerve growth factor receptors (NGFRs), vascular endothelial growth factor receptors (VEGFRs), platelet-derived growth factor receptors (PDGFRs), fibroblast growth factor receptors (FGFRs) and insulin receptors (InsRs) [43].

A series of consecutive reactions leads from ligand-bound receptor to Erk activation. Briefly, upon association with its ligand, the RTK dimerizes and trans-autophosphorylates on several tyrosine residues at its cellular domain [44]. The phosphorylated tyrosines serve as scaffolds for SH2- and PTB-containing cytoplasmic enzymes [38]. One of the protein complexes that bind to a phosphotyrosine on the RTK is Grb2-Sos, which in turn activates the small GTPase Ras. Active, GTP-bound, Ras recruits Raf proteins (A-Raf; B-Raf and c-Raf/Raf1) [45], the MAP3Ks of the Erk pathway. Raf kinases phosphorylate the MAP2Ks Mek1 and Mek2. Phosphorylated Meks dually phosphorylate Erk1/2 on neighboring Thr and Tyr residues, part of a TEY motif located at the activation loop. Several additional MAP3Ks may activate Mek, depending on the context of the cell and the type of stimulus (i.e., MOS [46], TPL2/Cot [47] and MLTK [48]). With only a few exceptions, Meks are the only known activators of Erk1/2. Without MEK-mediated dual phosphorylation, Erks are catalytically inactive. Erks can also be activated by GPCRs involving different subunits of G-proteins or β-arrestin, in a ligand-independent mechanism [49,50,51,52]. Thus, a variety of ligands, which activate either RTKs or GPCRs, as well as various environmental changes, lead to Erk activation. In addition to the interaction with the direct upstream activators, Erks interact with scaffold proteins such as KSR [53] and Mek partner-1 (MP-1) [54], which facilitate the association of the various cascade components thereby increasing the efficiency of their activation [55]. Activated Erks phosphorylate their substrates on Ser or Thr residues, in all cellular compartments. Cytoplasmic substrates include protein kinases, such as Rsk1/2, Mnk1/2 and Msk1 [56,57,58]. Nuclear targets include transcription factors of the Fos, Myc and Ets families [59]. Erks also phosphorylate upstream pathway components, such as Raf-1, B-Raf and Mek, as part of positive and negative feedback mechanisms [60,61,62,63,64]. Another manner of negative feedback is the Erk-induced expression of its own deactivating phosphatases [65]. For review on Erk substrates and downstream targets see [17].

#### 1.1.3. Erk Activation Is Achieved by Dual Phosphorylation of a TEY Motif within the Activation Loop

The difficulty in obtaining mutations that render Erks intrinsically active may stem in part from its tight regulation, supported by unique structure-function properties that distinguish them from most other eukaryotic protein kinases (ePKs). Almost all ePKs share a common kinase domain, which includes a highly conserved ATP binding site, a catalytic site and an activation loop. ePKs reside in equilibrium between active and inactive conformations, so that these catalytically-relevant sites are functional only in the active conformation. The kinase domain of all ePKs, including Erks, consists of a small N-lobe and a larger C-lobe (Figure 1A). The N-lobe contains 5 β-strands and a single helix (αC-helix), which is dynamic and occupies the space between the lobes. The αC-helix contains a conserved Glu, which, in the active conformation, forms a salt bridge with a Lys residue located within the AXK motif in β3 strand. This bridge is important and conserved in all ePKs and ensures anchoring and proper orientation of the ATP molecule. The C-lobe, which is mainly α-helical, binds and brings substrates adjacent to the ATP. A short (20–30 amino acids long) fragment located between the N- and C-lobes, known as the activation segment, contains some important elements, such as the DFG and APE motifs, the P+1 site, the catalytic and the activation loops (Figure 1A; [66]). The DFG motif is important for proper positioning of the ATP for phosphate transfer. While the Asp in the DFG is critical for recognizing the Mg^+2^ ions, the Phe forms hydrophobic interactions with the αC-helix and also with the catalytic Asp of the Y/HRD motif. The Y/HRD motif belongs to the catalytic loop responsible for catalysis. The conserved catalytic Asp of the Y/HRD motif functions by orienting the phosphate accepting hydroxyl as well as a proton-transfer acceptor. The Tyr/His residue of this motif, which is also conserved, is part of the R-spine and forms hydrophobic interactions with the DFG. An important conserved moiety is a phosphoacceptor (commonly a threonine) within the activation loop. In most ePKs, phosphorylation of this Thr is a pre-requisite for activity, as it is essential for shifting the equilibrium towards the active conformation. This phosphorylation induces several structural changes, including a conformational change of the DFG motif (to DFG ‘in’), rotation of the αC-helix, which enables the formation of a Glu–Lys salt bridge, and a domain closure between the N- and C-lobes, which ultimately stabilize the regulatory and the catalytic spines of the enzyme [67,68].

As this dramatic shift from the non-active to active conformation is a result of the single phosphorylation event, the activation-loop phosphoacceptor Thr is an obvious target for regulation and for mutagenesis aimed at generating activating variants. In several ePKs, converting this Thr to Glu resulted in a constitutive activation of the kinase. However, for most ePKs, genetic manipulations are not required to achieve constant activity, because these enzymes are capable of autoactivating in a mechanism that probably involves dimerization, which enforces a ‘prone-to-autophosphorylate’ conformation [69,70,71]. The rate of this spontaneous autophosphorylation is different in each ePK, but in the majority of cases it is sufficient to give rise to a significant activity [69]. In Erks, autoactivation is extremely inefficient and almost non-measurable [72,73]. Erks are fully dependent, therefore, on MAP2Ks for activation loop phosphorylation and induction of catalysis. The lack of autophosphorylation/autoactivation capability in Erk molecules makes overexpression a non-useful experimental approach for studying their biological and pathological effects. As the overall structure of the kinase domain of Erk is very similar to that of the other eEPKs (Figure 1), the explanation for the lack of autophosphorylation and basal catalytic activity of Erks is not trivial [69]. Not only that Erks are incapable of autophosphorylation as opposed to most ePKs, several other structural features also distinguish them from common ePKs. For example, although, like most ePKs, Erks seem to reside in equilibrium between two conformations (termed L and R; [74]) these conformations differ from the classical active and inactive conformations of ePKs. For example, no conformational change in the DFG motif (from ‘out’ to ‘in’) is apparent in the structure.

The differences in the biochemical properties between MAPKs and other ePKs may be associated with structural motifs, not part of the kinase domain, which are not present in other ePKs. Two such prominent motifs are the MAPK insert and the C-terminal extension, which includes a domain termed the L16 helix (Figure 1B). However, a bioinformatics-based evolutionary study suggests that the inability to autophosphorylate and the dependence on MEK may stem from minor structural differences, and not necessarily involving the MAPK insert, or the C-terminal extension [75]. This study reconstituted an inferred common ancestor of Erk1, Erk2 and Erk5 that is able to autophosphorylate and an ancestor of Erk1 and Erk2 that cannot. Analysis of the two ancestors suggested that a single amino acid deletion in the linker loop connecting the αC-helix and the β3 strand (position 74 in modern Erk1) and a mutation in the gatekeeper residue (Gln122 in modern Erk1) account for the loss of autophosphorylation and dependence of modern Erk on its upstream activator. Indeed, inserting these two modifications into modern Erk1 was sufficient to generate Erk1 molecules that, when tested in kinase assays in vitro, showed high autophosphorylation ability and consequently catalytic capabilities similar to those of Mek-phosphorylated Erk1 [75]. This study clearly points at residues and domains that could be manipulated in an effort to generate intrinsically active, Mek-independent, Erks. These same residues were identified, in fact, as candidates for mutagenesis by other approaches as well [76]. 

Other unique Erk domains are the substrate binding motifs. Erks possess two distinct sites through which substrates, activators and deactivators can bind. The first is the common docking (CD) site, which is located about 10Å from the active site of Erk2 (Figure 1B) and is composed of amino acids such as Asp316 and Asp319. The second docking site of Erk2 is the hydrophobic DEF pocket, which is composed of residues Met197, Leu198, Tyr231, Leu232, Leu235 and Tyr261, and exists only in dually phosphorylated Erk adjacent to the catalytic site (Figure 1C). Important mutations, discussed here, occurred in these two domains [73,77,78].

#### 1.1.4. ERK Molecules Are Highly Active in Most Cancers, but Oncogenic Mutations in ERK Themselves Are Very Rare

All upstream components of the Erk signaling cascade are frequently mutated in cancer [79], and it is believed, therefore, that Erks are abnormally overactive in essentially all cancer cases [80]. Accordingly, the Ras-Raf-MEK-Erk cascade has become a major target for anti-cancer therapy [81]. Oncogenic mutations or other genetic alterations (e.g., gene amplification) have been found in RTKs, Ras, Raf and MEKs, but no activating mutations or genetic alterations in Erk molecules themselves have been reported as oncogenic in tumor viruses or in patients. However, as the only known substrates of Rafs are the MEKs, and the only known substrates of MEKs are the Erks, it implies that the biological and pathological/oncogenic effects of the pathway are mediated exclusively via the Erk proteins (note some reports on deviations from the linearity of the RTK-Ras-Raf-MEK-Erk tier: [41,49,50,51], reviewed in [17]). It is not clear, therefore, why mutations in Erks are rarely found in cancer patients. This situation could be taken as another indication for the unusual tight regulation bestowed on Erks by the specific structural motifs, immunizing them against mutations that render them spontaneously active and oncogenic. Indeed, although some mutations that render Erks intrinsically active and even oncogenic (one mutation) have been discovered in the laboratory, they do not activate Erk to the maximal levels possible (that of Mek-activated Erk), and their oncogenic effect is markedly weaker than that of active, oncogenic, Ras or Raf [82,83].

Nevertheless, mutations in ERKs have been identified in a small number of patients (Table 1) and at least one of those, E320K in ERK2, seems to appear in a few dozen patients suffering from cervical and head and neck carcinoma (Table 1B). Perhaps activating mutations in ERKs are not fully oncogenic and do not have a causative effect, but may promote the disease.

#### 1.1.5. Erk Inhibitors Have Been Recently Developed

Not only that Erks are obvious targets for anti-cancer therapy because they are the downstream components of the RTK-Ras-Raf-MEK pathway, in the majority of cases of tumors resistant to EGFR, B-Raf and MEK inhibitors, re-activation of Erk is observed [84]. These findings reinforce the need for direct Erk inhibitors. Specific inhibition of Erks should also be a powerful tool for research. For unknown reasons, developing pharmacological Erk inhibitors has lagged behind the development of inhibitors against the other MAP kinases, JNK and p38 and has required unusual efforts. Morris et al., for example, screened approximately five million compounds and performed multiple improvement steps in order to discover SCH772984, a small molecule that inhibits both Erk isoforms with an IC_50_ at the nano-molar range [85]. Further development of this inhibitor provided an orally administered analog, MK-8353 [86], which is being tested in phase-I clinical trials. Yet another potent Erk inhibitor is BVD-523 (Ulixertinib), a selective and reversible Erk1 and Erk2 ATP competitive molecule [87], which is currently in phase II clinical trials. GDC-0994 (Ravoxertinib) [88], a pyrazolylpyrrole-based inhibitor that was optimized specifically towards Erk by using structure-guided methods [89], is also undergoing clinical testing. Additional compounds that exhibit selectivity towards Erk are LY3214996 [90], FR180204 [91], VRT-11E [92] and the Erk dimerization inhibitor DEL22379 [93]. For a comprehensive description of Erk inhibitors, see [94].

In parallel to the biochemical and pharmacological characterization of the inhibitors, a large number of mutations that render Erk proteins resistant to them have been reported [95,96,97] (Table 1).

## 2. Identification of Various Mutations in ERKs and Study of Their Properties

### 2.1. Almost All Known Mutations in ERKs Have Been Identified Experimentally

Unlike the many mutations known in RTKs, Ras, Raf and MEK, mostly identified in tumors, almost all mutations known in ERKs have been identified in laboratory setups. Only a few have been identified in cancer patients, and even for those, it is not clear whether they are associated with the disease. Experimental systems were initially designed for identification of intrinsically active Erks. Later, following the development of Erk inhibitors, genetic screens were developed for identification of mutations that would cause drug resistance. The mutations identified in patients, in screens for drug-resistant molecules and for intrinsically active Erks are summarized in Table 1. It should be noted that numeration of the mutations in ERK1 (in text and in Table 1A) refer to the sequence of the human protein and numeration of mutations in ERK2 (in text and in Table 1B) refer to the sequence of rat protein. Notably, mutations that had been discovered (until 2006) in ERK orthologs in lower organisms were summarized in [98] and will not be discussed here.

### 2.2. Mutations Produced on the Basis of Structure-Function Studies

Original attempts to develop intrinsically active Erk molecules took the conventional approach of trying to mimic the activatory phosphorylation of the activation loop. As no phosphomimetic residue is available for tyrosine, this approach was limited to modifying the Thr of the TEY motif and turned out to be ineffective [99,100]. In fact, not only was changing this Thr to Glu unsuccessful, but the resulting Erk2^T183E^ enzyme showed lower activity than Erk2^WT^, even when phosphorylated by MEK [98,101,102]. Furthermore, even when mutations that render Erk2 intrinsically active were discovered, combining them with the T183E mutation did not create a more active molecule [103].

Other mutations devised on the basis of structure-function understanding, however, did lead to the development of interesting mutants. For example, mutating the gatekeeper residue of Erk2 resulted in an intrinsically active variant (Q103G and Q103A) [76]. Furthermore, mutating residues that interact with Gln103 provided even more active variants (when tested as purified recombinant proteins), primarily Erk2^I82A^ and Erk2^I84A^ [76]. As described above, the gatekeeper residue was also discovered as a site that distinguishes between an inferred ancestor kinase, which is capable of autophosphorylation, and the modern Erk1 and Erk2. Mutating the gatekeeper residue in Erk1 on the basis of comparison to the inferred ancestor (inserting the Q122M mutation) rendered the mutant intrinsically active [75].

The mechanism that renders all these mutants intrinsically active was shown to be the acquisition of an autophosphorylation capability. Namely, the mutations did not impose adoption of the native conformation, but rather unleashed an obstructed autophosphorylation capability and allowed autoactivation. These observations suggest that Erks are similar to most other ePKs that possess the autophosphorylation machinery, but this activity in Erks is not spontaneous. Structural blockers of autophosphorylation in Erks are not known implying that activating mutations could reveal them. An interesting mechanism of action for how substitutions at Gln103 or Ile84 unblock autophosphorylation was suggested by Emrick et al. It was proposed that the mutations induce a pathway of intramolecular interactions leading to flexibility in the activation lip, thereby enabling the phosphoacceptors to reach the catalytic Asp (D147 in Erk2). In the inactive form of Erk2, the intramolecular pathway includes hydrophobic interaction between Leu73, Gln103 or Ile84 and phe166 of the DFG motif, which in turn is linked to L168 through the backbone. The latter forms side-chain interactions with Val186 from the activation lip. Emrick et al. further suggested that T188 forms hydrogen bond with Asp147 and Lys149. These interactions together impede autoactivation by holding the activation lip in a stable conformation. It is therefore not surprising that mutating Leu73, Gln103 or Ile84 would lead to the movement of Phe166 and thus affect the hydrophobic interaction between Leu168 and Val186 and the hydrogen bond between Thr188 and Asp147/Lys149. This would cause an increase in flexibility of the activation loop and eventually autophosphorylation.

Although, when tested in vitro as recombinant proteins, these mutations render the Erk molecules intrinsically active, the significance of the mutants in the gatekeeper and nearby residues in living cells is not clear. While Erk2^I84A^ seems to be spontaneously active when expressed in HEK293 cells, it is not spontaneously active in NIH3T3 cells. The equivalent Erk1 mutant, Erk1^I103A^ is not spontaneously active in either cell line [82]. None of the mutants can oncogenically transform NIH3T3 cells [82], but, intriguingly, several mutations in the gatekeeper of Erk2 (Q103) were identified in screens for drug-resistant Erk2 molecules (Table 1B) and a mutation in I82 (I82T) was found in one cancer patient (Table 1B). Perhaps conversion of these residues to particular amino acids (e.g., Thr), other than those tested so far (i.e., Ala) would render them more active in cells and possibly even oncogenic.

Another mutation that was generated in Erk2 on the basis of structural studies is S151D. This mutation was designed following an alignment of the conserved sequence DLKPSN in MKK1 with the sequence DLKPEN in cAMP-dependent protein kinase. This mutant resulted in a 15-fold enhancement of MKK1 activity [99] and was therefore attempted in Erk2 [103]. Erk2^S151D^ manifested MEK-independent activity that was about 15-fold higher than that of Erk2^WT^, but was just 1.5% of that of MEK-phosphorylated Erk2^WT^ [73,103].

### 2.3. Only a Few of the Mutations Identified via Genetic Screens in ERK Orthologs of Evolutionarily Low Organisms Are Relevant to Mammalian Erks

High throughput screens in *S. cerevisiae* and *D. melanogaster*, provided a variety of gain-of-function mutants of Erk orthologs in these organisms [73,77,104,105] (Reviewed in [98]). These mutations allowed important insights into the modes of regulation of the given Erk ortholog in each case and to interesting cross talks between yeast MAPK pathways. The relevance of most of those to mammalian Erks is, however, unclear, because some of the mutations occurred in residues that are not conserved in the mammalian enzymes, and of the conserved residues only a handful were tested [73,103,106]. Overall, very few of the mutations turned out to be relevant to mammalian Erks, but these are of significant importance.

For example, insertion of the L73P mutation to Erk2 (equivalent to the L63P mutation in Fus3 [107]) rendered Erk2 intrinsically active, as tested by an in vitro kinase assay with purified recombinant proteins, but to an activity level of approximately 1% of the activity displayed by Mek-activated Erk2 [103]. Combining L73P with other mutations, such as S151D and D319N, created a more active enzyme [103]. It required a combination of three mutations, L73P+S151D+D319N to create an Erk2 protein with a MEK-independent activity that was 100-fold higher than that of wild type Erk2 in vitro. Notably however, this activity is just about 6% of the MEK-phosphorylated Erk2 activity [103]. Thus, these mutants are *bona fide* intrinsically active, but their activity is not very high. Interestingly Leu73 is part of the hydrophobic cluster affected by the “gatekeeper mutations”. It seems that mutations in Ser151 also interfere with the contacts of the catalytic base Asp147 with Thr188, resulting in increased activation lip flexibility and activation of the phosphoacceptors Tyr185 and Thr183.

The only Erk mutant that has been shown so far to oncogenically transform cells in cultures, Erk1^R84S^, was generated on the basis of a mutation in the yeast ortholog Mpk1/Slt2. The mutation in Mpk1/Slt2, R68S, was identified in a screen that looked for Mpk1 mutants that rescue the phenotype of cells lacking the relevant MEKs [73]. Six Mpk1 mutants were isolated, but only the R68S mutation was found relevant to Erks of higher eukaryotes, including of *Drosophila* and mammals [73]. *Drosophila* and mammalian Erks carrying the equivalent mutation (R80S in *Drosophila’s* ERK/Rolled; R84S in mammalian Erk1; R65S in mammalian Erk2) displayed high spontaneous intrinsic catalytic activity (>30% of the activity of Mek-activated Erk), independent of Mek activation in in vitro assays [73,82], in cell cultures [82] and in vivo, in transgenic mice and flies [83,108]. Furthermore, Erk1^R84S^ and Rolled^R80S^ were shown to function as oncogenes, capable of transforming NIH3T3 cells and to give rise to tumors in the fly, respectively [82,83]. Erk1^R84S^ was also shown to cause mild cardiac hypertrophy, when expressed as a transgene in the heart of mice [108]. The basic mechanism of action of the R80S/R84S/R65S mutation is similar to that of the other intrinsically active variants described above, namely, it bestowed upon the Rolled and Erk proteins an efficient autophosphorylation capability [73,82,83]. The structural basis for this capability is not clear, in part due to the extreme flexibility of Arg65 within the Erk2 structure. In many Erk2 structures it accommodates a different conformation (Figure 2).

Arg65 is located at a pivotal position in the αC-helix, the conserved helix within the N lobe, and interacts with the L16 domain, which is flexible and unstructured in the inactive form. Interestingly, in spite of the different conformation adopted by Arg65 in the various crystal structures of Erk2 (Figure 2), in many of them Arg65 is in association with amino acids of the conserved DFG motif (D165, F166, G167). In the structure of Erk2^WT^, PDB 1ERK, or 5UMO, Arg65 seems to have two possible conformations so that it forms a hydrogen bond with either the side chain of Asp165 or the backbone of Gly167. In the structure of Erk2^R65S^ (determined at a high resolution of 1.48Å (PDB 4SZ2)) the substituted serine is smaller than arginine and is not in association with the DFG motif. Instead, a new hydrogen bond is formed between Ser65 and Tyr34 of the P-loop (see Figure 8 in reference [82]). In addition, Ser65 stabilizes Thr183 from the activation loop and it is hypothesized that the Asp165 of the DFG motif can interact with ATP. It is noteworthy that in the crystal structure of dually phosphorylated Erk2 (PDB 2ERK), Thr183 interacts with the αC-helix, particularly with Arg65 via a water molecule [109].

Interestingly, similar to the case of Erk2^R65S^, in the crystal structure of the intrinsically active Erk2^I84A^ (PDB code 4S30), a mutant that also autophosphorylates efficiently, the interaction of Arg65 with the DFG is also abolished, although the mutated residue is located in a distance from the αC-helix. As discussed above, the I84A mutation affects a hydrophobic cluster, involving, amongst other residues, Leu73 of the αC-helix, which may in turn divert Arg65. Also, in the Erk2^I84A^ structure in complex with AMP-PNP (PDB 4S34), a shift of Tyr34, causes it to form a hydrogen bond with Thr66 in addition to the Pi-Pi interactions with Tyr62 of the αC-helix observed in Erk2^WT^. Tyr34 in Erk2 plays a pivotal role in catalysis and the DFG, especially Asp165, is involved in ATP binding by interacting with the gamma phosphate of ATP. Erk2 bearing mutations in Tyr34 (Y34H/N, in ERK1: Y53H) or Tyr62 (Y62N, in ERK1: Y81C) were shown to acquire resistance to the Erk inhibitors SCH77984 and VRT-11E in both ERK1 and ERK2 in two different screens (see below [95,96]). Finally, the association between Arg65 of Erk2 and the DFG is also abolished in other intrinsically active mutants, including mutations found in the CD site (PDB 6OT6) [110].

It is thus conceivable that the association of Arg65 with the DFG motif is crucial for blocking autophosphorylation activity by acting as a barrier between the ATP binding pocket, DFG motif and the activation loop. Tyr34 and Tyr62 of the P loop, which slightly change their conformation in several active mutants, may also play some role in suppressing spontaneous activity. Notably, mutations in Arg84 of Erk1, equivalent to Arg65 in Erk2, were identified in two cancer patients (Table 1A).

Intriguingly, another gain-of-function mutation, Y268C [73], was identified in the same genetic screen in yeast that provided the R68S mutation in Mpk1/Slt2. It was later shown that Y268A is also a gain-of-function mutation in Mpk1 [78]. Tyr268 is located at the heart of the DEF pocket, and in mammalian Erk2 a mutation in the equivalent Tyr, Y261A, is a partial loss-of-function mutation because Erk2^Y261A^ cannot bind and phosphorylate some of its substrates and cannot execute some of its biological functions [29,111,112]. As the mechanism through which Mpk1^Y268A/C^ function as gain-of-function mutants is unknown, it is also unexplained why the equivalent mutations in Erk2 cause a loss of function [78].

Similar to the mammalian Erks, Rolled, the Erk ortholog in *Drosophila melanogaster*, is involved in numerous processes in the fly, including the development of the eye [23,24]. A screen aimed at isolation of mutants that facilitate proper eye development even in the absence of the ligand that activates the Erk pathway in the eye, identified a mutant fly that was termed *sevenmaker*. The mutation it carried was found to be D334N in the *Drosophila’s* ERK/Rolled [77]. An equivalent mutation was isolated in another screen in yeast designed for identifying Fus3 molecules that are not inhibited by Hog1 [107], in a large-scale screen performed in mammalian cells for gain-of-function and inhibitor-resistant Erk2 mutants (D319 in Erk2) [95], and in cancer patients. As the *sevenmaker* mutation occurs at the CD site, its proposed mechanism of action is an elevated resistance to MAPK phosphatases resulting in increased sensitivity to low levels of MEK’s activity [106]. This notion was based on the observation that the Erk2^D319N^ protein was less susceptible to inactivation by the MAPK phosphatase CL100 than the Erk2^WT^ recombinant protein, while both undergo inactivation in a dose-dependent manner [106]. Inserting this mutation into mammalian Erk2 showed minimal effect on catalytic activity in in vitro experiments and in cell culture [73,103,106]. It is not clear how the *sevenmaker* mutation affects Erk’s association with phosphatases, but not with activators and substrates that also utilize the CD site. Misiura et al. have shown, quantitatively, that the *sevenmaker* mutation increases the catalytic activity of Erk by changing the interaction energies. It specifically modifies the enzyme’s susceptibility to deactivation by phosphatases, while not affecting the activation process by the MAPK kinase [113]. Nonetheless, the *sevenmaker* mutation may also affect catalysis *per se*. Indeed, recombinant Erk2^D319N^ does not manifest unusual catalytic properties, but when combining the D319N mutation with an activating mutation there is a dramatic elevation of catalysis [73,83,103]. Since Erk phosphatases do not exist in in vitro assays, the effect of the *sevenmaker* mutation should be explained by other mechanisms. Probably, in addition to reducing the affinity to phosphatases, the *sevenmaker* mutation also confers a conformational change that further stabilizes the “prone-to-autophosphorylate” conformation [69] induced by another mutation on the same protein. Supporting the role of the *sevenmaker* residue not only in substrate binding, but also in activation of catalysis, Molecular Dynamic simulations suggest that stabilization of the active conformation of Erk2, following phosphorylation of Thr183, is associated with disruption of several hydrogen bonding involving Asp334 [114]. Furthermore, in the structure of Erk2^D319N^ the interaction of Arg65 with the DFG is lost, a property of several of the autophosphorylating Erk mutants. Aside from this disruption, the crystal structure of Erk2^D319N^ is, essentially, indistinguishable from that of Erk2^WT^ [110].

The *sevenmaker* site seems to be a hot spot for mutations as it is being re-discovered in different screens. Mutations in Asp319 of Erk2 (D319N, D319V) or mutations in the neighboring residue (E320K and E320V), as well as in Glu79, which associates with D319, (E79K), were identified in a comprehensive screen that searched for gain-of-function and drug-resistant mutations, using A375 cells (see below; [95]). The *sevenmaker* mutation, itself, D319N, was also reported in four cases of carcinoma (COSMIC ID: D319N in ERK2—COSM98175) and three other patients carried another substitution in the 319 position (Table 1B). Interestingly, the ERK1 *sevenmaker* site was not found to be mutated in any screen, or in cancer patients.

### 2.4. Mutations in ERKs Are Very Rare in Cancer Patients, but Some Are Similar to Those Identified in Laboratory Models

Although several dozens of mutations in ERK1 and ERK2 have been identified in cancer patients (Table 1) the rate of mutations in ERK in patients is very low and most of the mutations appeared in only one of the samples tested. Mutations in ERK2 were observed in 179 out of 60,712 unique samples, approximately 0.3%. The COSMIC database lists 148 reported ERK1 mutations, out of 47,784 tumor samples tested (also approximately 0.3%). It is not clear whether these mutations have any causative effect on the malady. An exception is the E320K mutation, which was observed in 27 patients of squamous cell carcinoma (COSMIC ID: E320K in ERK2—COSM461148). Mutations in Glu320 were also identified in a screen for drug resistant Erk molecules. Notably Glu320 is neighboring Asp319 (the location of the *sevenmaker* mutation) within the CD site. Just like the *sevenmaker* mutation, the E320K does not affect the enzyme’s intrinsic catalytic properties as tested in vitro with recombinant Erk2^E320K^, and in transient transfections of HEK293 and NIH3T3 cells ([110]; Smorodinsky-Atias and Engelberg, unpublished observation). It may function, therefore, similar to the mutations in Asp319 by reducing the protein association with phosphatases (also supported by Brenan et al. [95]). Yet, unlike D319N, the E320K mutation enforces significant structural changes on the crystal structure of Erk2 and on its biophysical properties. Also, when equivalent mutations were inserted to the *Drosophila ERK/Rolled* they conferred different properties on the protein, suggesting that D319N and E320K function via different mechanisms [110,115]. More mutations of note identified in patients occurred in Glu79 (COSMIC ID: in ERK2—COSM444794), Ser140 (COSMIC ID: in ERK2—COSM3552430) and Pro56 (COSMIC ID: in ERK2—COSM4471756) residues. These substitutions were subsequently discovered as gain-of-function mutations in the screen that tested all possible missense mutations of ERK2 [95] (see below).

Nearly all of the ERK1 mutations recorded in patients, appeared only once. Only 18 mutations occurred in two or three independent samples. Notably, two patients carried the R84H mutation in ERK1. As discussed above, another mutation at the same location, R84S, was shown to render Erk1 capable to oncogenically transform cells in culture [82]. This finding calls for testing the oncogenic potential of Erk1^R84H^ and perhaps of more Erk1 molecules in which Arg84 was substituted.

### 2.5. A Large Number of Mutations Can Render Erks Resistant to Pharmacological Inhibitors

The development of specific inhibitors towards Erks was the impetus for a series of studies that searched for mutations that cause drug resistance. Goetz et al., constructed a library of randomly mutagenized ERK1 and ERK2 cDNAs and induced its expression in A375 melanoma cells (harboring the BRAF^V660E^ oncogene) in the presence of either the Erk inhibitor VRT-11E, the MEK inhibitor trametinib, or with a combination of trametinib and the Raf inhibitor dabrafenib [96]. Overall, sequencing the ERK1 or ERK2 molecules in cell populations that survived the treatment identified 33 mutations in ERK1 (in 28 amino acids) and 24 in ERK2 (in 20 amino acids). In a separate screen for A375 colonies resistant to VRT-11E, another five mutations in ERK2 were discovered. All mutations are presented in Table 1. Only five of the mutations that caused resistance to VRT-11E were identified in both isoforms. These were (in ERK1/ERK2 order) Y53H/Y34H/N, G54A/G35S, P75L/P56L, Y81C/Y62N and C82Y/C63Y. A mutation at only one of the residues that caused resistance to the MEK inhibitor was also common in the two isoforms, Y148H/Y129N/H/F/C/S. Importantly, some of the residues that were found mutated in this screen were also reported to be mutated in patients, including Arg84 and Gly186 in Erk1 and Asp319 and Glu320 in Erk2. Another important finding of this study, with significant implications to therapeutic strategies, is that Erk variants that are resistant to RAF/MEK inhibitors are sensitive to Erk inhibition and vice versa [96].

As the mutations were not tested on purified Erk proteins it is not known how they affect the intrinsic biochemical and structural properties of the enzymes leaving the detailed mechanism of the acquired drug resistance open for future studies. It is possible however that mutations identified in the Erk-inhibitor screen interfere with drug binding as they cluster in proximity to the ATP/drug binding pocket. These mutations are located in the glycine-rich loop and the loop between β3 and αC-helix (in ERK1: I48N, Y53H, G54A, S74G, P75L. In ERK2: Y34H, G71S, P56L). Mutations residing in the αC-helix itself (ERK1: Y81C, C82Y, ERK2: Y62N, C63Y) may function similarly. ERK1 mutants that were identified in the Raf/MEK inhibitors screen probably function via a different mechanism. They are distributed along the molecule, but seem to cluster in domains important for catalysis and may render the kinase catalytically active. A206V/A187V and S219P/S200P (of ERK1/ERK2), for example, reside in the activation lip, R84H, C82Y and Q90R of Erk1 map to the αC-helix and Y148H of Erk1 and Y129F, D319G and E320K of Erk2 are located in the CD site. Mutations in the *sevenmaker* residue were already discussed above and probably function by reducing affinity to phosphatases in combination with some effect on catalysis. Another mutation, R84H in Erk1, occurred at the same residue in which the only oncogenic mutation identified so far in Erks, R84S, also occurred [82]. As Erk1^R84S^ was shown to be intrinsically catalytically active and spontaneously active when expressed in culture cells and in transgenic mice [82,108], it could be that Erk1^R84H^ also acquired similar properties and is independent of upstream activation, making it resistant to Raf and MEK inhibitors. Indeed, kinase assays using the various mutants, expressed in and immunoprecipitated from A375 cells, showed that the mutants maintained activity in the presence of either VRT-11E or SCH772984, another Erk inhibitor [96]. On the basis of the immunoprecipitation kinase assay it could be suggested that the mutants arising from the screen in the presence of RAF/MEK inhibitors acquired the capability to maintain sufficient kinase activity, even under these conditions, thereby rescuing the transformed cells from the inhibitors. As some of these mutations are found in patients, this conclusion is therapeutically relevant and may suggest that RAF and MEK inhibitors should be contraindicated for patients that harbor these mutants in the tumor.

Intriguingly, the mechanisms of action of the various mutants in vivo may be different for Erk1 and Erk2 as some of the mutations seem to be isoform-specific. Mutations in αC-helix were found only in ERK1 (C82Y, R84H and Q90R), while mutations in the CD site, such as D319G and E320K were only found in Erk2. Although the differences in the occurrence of the mutations could be a result of an unsaturated screen, the notion that the mutations are isoform-specific is supported by the situations observed in patients. The E320K mutation of Erk2 was found in 27 cancer patients while the equivalent mutation in Erk1 was not reported. Similarly, the *sevenmaker* site of Erk2, D319, which was mutated in seven patients, was not found so far to be mutated in ERK1 in patients. In line with these observations Goetz et al. inserted the equivalents of the D319G and E320K mutations into ERK1, and observed that the resulting proteins, Erk1^D338N^ and Erk1^E339K^ were not resistant to inhibitors, at least in the cell assay [96]. It is reasonable to conclude that different mutations may render Erk1 and Erk2 resistant to drugs. Given that the biological functions of the two isoforms is almost fully redundant [36] and that the inhibitors affect both isoforms similarly *in vitro*, it is currently difficult to explain the dichotomy in the mutations that cause resistance of each isoform.

A mutation that renders Erks resistant to SCH772984 was identified when cells of the colorectal cancer cell line HCT-116 (harboring a mutated KRAS) were serially passaged in the presence of increasing concentration of the inhibitor for 4 months [97]. Sequencing the ERK1/2 genes isolated from resistant clones, revealed a reoccurring point mutation, G186D, in ERK1 [97]. Gly186 resides in the DFG motif of the activation segment. The mutant displayed a several-fold reduction in binding affinity to the inhibitor, compared to the wild-type protein. Crystal structure of SCH772984-associated Erk2 suggests that this reduction is a direct result of a steric clash imposed by the aspartic acid in the active site, which destabilized the binding of the inhibitor. The same Erk1^G186D^ mutant did not provide resistance to another ATP-competitive Erk inhibitor (VRT-11E), an observation that is explained by the structural difference between the two inhibitors, significantly altering the interactions with the binding pocket, predominantly the distance of the molecule from the new aspartic acid [97]. Notably, the orthologous residue in Erk2, Gly167 was found mutated to Asp in a screen for Erk2 mutants that are resistant to SCH772984 and VRT-11E in A375 cells. In this screen Gly167 was mutated to many other residues, but only the Erk2^G167D^ was selected as rendering the kinase drug-resistance [95]. The interference of Asp at position 169 for drug binding seems very particular.

Erk2^G167D^ was in fact isolated in a large-scale screen that led to the discovery of many more mutants. In this effort, Brenan et al. employed saturation mutagenesis and were able to screen a library of 6810 variants of ERK2, out of 6821 possible, each carrying a point mutation. They searched for gain-of-function, loss-of-function, as well as for drug-resistant mutants. The ERK2 mutants library was introduced into A375 cells and inducibly expressed, under the premise that cells expressing gain-of-function mutants will proliferate slower, while cells expressing loss-of-function mutants will proliferate faster, than cells expressing Erk2^WT^ [95]. The relative abundance of Erk2 molecules that were expressed in the cells after 96 h was determined by parallel sequencing.

Indeed, mutants considered to be carrying GOF mutations on the basis of previous screens or appearance in tumors, such as Glu320 and Asp319, were depleted from the proliferative culture. Mutations in an additional 19 residues caused the Erk2 molecules to be depleted to a degree equivalent to or greater than that of Erk2^Glu320^ and Erk2^Asp319^. The 19 residues were Glu79, Gly83, Gly8, Leu333, Pro56, Val47, Ser358, Val16, Pro354, Arg13, Glu320, Glu58, Ala3, Ala350, Ala7, Ser140, Thr24, Gln15, Asp319, Gly20 and Phe346 (from the most depleted to the least). Many of the mutants considered to have acquired gain-of-function mutations were shown to be catalytically active when immunoprecipitated from cells treated with the RAF inhibitor trametinib. As the mutants were not tested as recombinant purified proteins it is not clear whether they possess any unusual catalytic properties, and, specifically, if they are intrinsically active.

The mutants library was also inducibly expressed in A375 cells exposed to sublethal doses of VRT-11E, or SCH772984 in order to discover Erk2 mutants that are resistant to these inhibitors. The relative enrichment of the Erk2 mutants was quantified after 12 days of exposure to inhibitors. The rationale in this experiment was that cells harboring an Erk2 mutant insensitive to an inhibitor would allow proliferation in its presence. Mutations in 12 residues rendered Erk2 resistant to VRT-11E (Arg13, Ile29, Gly30, Ala33, Met36, Val37, Arg65, Gln95, Met96, Asp98, Thr108, Leu154), mutations in 9 residues confer resistance to SCH772984 (Glu31, Tyr41, Val47, Lys53, Glu68, Leu67, Ile101, Asp122, Asp125) and mutations in 18 residues render Erk2 resistant to both inhibitors (Tyr34, Gly35, Cys38, Asp42, Val49, Ile54, Ser55, Pro56, Phe57, Gln60, Tyr62, Cys63, Thr66, Glu69, Leu73, Gln103, Gly167, Ile345). Curiously, half of the mutations that cause resistant to VRT-11E occurred in residues that make direct contacts with the inhibitor, but none of the mutants that confer specific resistance to SCH772984 are mutated in residues that contact the inhibitor. Projection of the mutations identified in this study into cancer associated mutations showed that within the top 20 residues harboring GOF mutations, five were reported to be mutated in patients (E320K/V, D319N/V, E79K, P56L and S140L). Erk2 mutants carrying any of these GOF mutations were found to rescue A375 cells from the anti-proliferative effect of Raf/MEK inhibitors, but only one, Erk2^P56L/G^ was able to rescue the cells from SCH772984. All GOF mutations clustered within the CD site, whereas LOF mutations altered the DEF pocket. The LOF mutations in the DEF pocket seem dominant since when combined, on a single Erk2 molecule, with a GOF mutation in the CD site the resulting protein was not able to rescue cells and promote the downstream signaling. It seems that the mechanism of action of the GOF mutants residing in the CD site is similar to that of the *sevenmaker* mutation. This hypothesis was validated by co-expressing Erk2 mutants with BRAF^V600E^ and dual specificity phosphatase (DUSP) in 293T cells and monitoring Erk2 phosphorylation. Erk2 molecules mutated at the CD site (E320K/V; D319N/V E79K) exhibited sustained phosphorylation levels relative to Erk2^WT^ [95]. It is worth noting that the GOF mutant P56L showed a reduced level of phosphorylation, suggesting that constant phosphorylation in the presence of DUSP is not a general property of all GOF mutations. Most mutants identified in this comprehensive study await further biochemical, structural and pharmacological analysis.

## 3. Discussion

Table 1 lists an impressive number of mutations in Erk1 and Erk2, many of them discovered in screens for drug-resistant mutants. The mutations that confer drug resistance outscore the very few mutations that render Erks intrinsically active and the single mutant that was shown so far to possess oncogenic properties. The mutations causing drug-resistant occur throughout the protein so that various mechanisms are involved. It is not clear whether the list in Table 1 reflects all relevant mutations possible in Erks. Indeed, most of the screens that provided the currently known mutations were high throughput, but they were performed in a particular context of a given cell line, or against specific pharmacological inhibitors. Also, it is not clear if these screens themselves were saturated. It is a plausible assumption therefore that further screening, particularly in novel experimental setups, would result in yet unidentified mutations. Also, once Erk inhibitors reach the clinic, patients may develop resistance by acquiring mutations other than those discovered in the laboratory. As the research of Erk mutations is relatively young many properties of the known mutation are still elusive. For most of the mutations that cause resistant to inhibitors the mechanism of action is not known and for many of them even the effects on Erks’ conformation and catalytic properties are yet to be revealed. Finally, it may take a long time to reveal the role (if any) of the rare mutations identified in cancer patients, in disease etiology. Most mutations that were identified in a single patient, may be bystanders with no role in the disease, while others may contribute to disease development or even disease onset. Prime suspects for a causative role are the R84H mutation in Erk1, because an equivalent mutation, R84S, was shown to be oncogenic in cells in culture, and E320K in Erk2 that is found in patients in a higher rate than all other mutations.

### 3.1. Mutations Identified in Erks Could Not Have Been Predicted by Structural or Mechanistic Analysis

It is not currently possible to predict the location and type of more Erk mutations. Although Erk proteins have been the subject of comprehensive structural studies, including via X-ray crystallography, NMR and HX-MS approaches, and although critical features of their structure-function properties have been revealed, it is not clear how to translate this knowledge into predicting specific mutations that will modify Erks’ biochemical or pharmacological properties or biological functions. Also, modifying Erks’ biological effects requires understanding of additional mechanisms, responsible for sub-cellular localization and interaction with partners and scaffold proteins. Some of these mechanisms could be isoform-specific and, consequently, mutations that affect these processes may be different in Erk1 and Erk2, similar to the case of the CD site mutations that seem relevant only to Erk2 and mutation in the αC-helix, which are more relevant to Erk1. As a result of our inability to translate structural and mechanistic knowledge into mutation design, most mutations identified so far were discovered via unbiased screens, and even following their isolation, their mechanism of action is not understood.

### 3.2. Mutations Identified in Erks Disclose 3 Hotspots for Mutagenesis, Perhaps Reflecting Some Prevailing Mechanisms

Although mutations discussed in this review are spread along the Erk molecules, some hotspots re-appear in several laboratory screens as well as in cancer patients. The *sevenmaker* residue and its neighbor at the CD site, Glu320, are prominent examples, relevant for Erk2. αC-helix, mainly Arg84/Arg65 (in ERK1/ERK2), is another hotspot, which seems relevant primarily to Erk1. Yet another important residue is the gatekeeper that was discovered as a target for mutagenesis by studying the evolution of Erks and by structural approaches. Mutations that occur in the CD site affect protein-protein interactions, primarily with phosphatases, while mutations in the gatekeeper, in the residues that are in proximity to it and in residues of the αC-helix, significantly increase the commonly negligible intrinsic catalytic activity of the kinases. All mutations that caused elevation of the basal catalytic activity did so via identical mechanism, increasing the autocatalytic capability of the proteins. So far, no mutation was discovered that allows Erks to bypass the requirement of activation loop phosphorylation and enforces by itself adoption of the active conformation. Rather, the mutations discovered so far caused the Erk molecule to acquire a ‘prone to autophosphorylate’ conformation. Erks may be immune against mutations that induce the active conformation by themselves.

## 4. Conclusions

### Clinical and Biochemical Lessons from the Erk Mutant

Most of the mutations described in this review seem to be directly relevant to understanding cancer etiology and patients response to drugs. An important lesson is that Erks, at least Erk1, could become oncogenic. This finding strongly suggests that the oncogenicity of the RTK-Ras-Raf-MEK pathway is mediated primarily via Erks, reinforcing the effort to inhibit Erk as a powerful anti-cancer approach. The analysis of the mutants may further suggest that isoform-specific inhibitors should be developed with higher priority to Erk1-specific inhibitors. This would be a difficult challenge for drug developers. The mutations that cause drug resistance could be already taken into consideration when a therapeutic strategy is planned. Namely, drugs should be applied according to the mutation that appears in the tumor. This requires deep understanding of the effect of each of the many mutations discovered so far (Table 1) on Erks’ biochemistry, biology and pathology.

## Figures and Tables

**Figure 1 cells-09-00129-f001:**
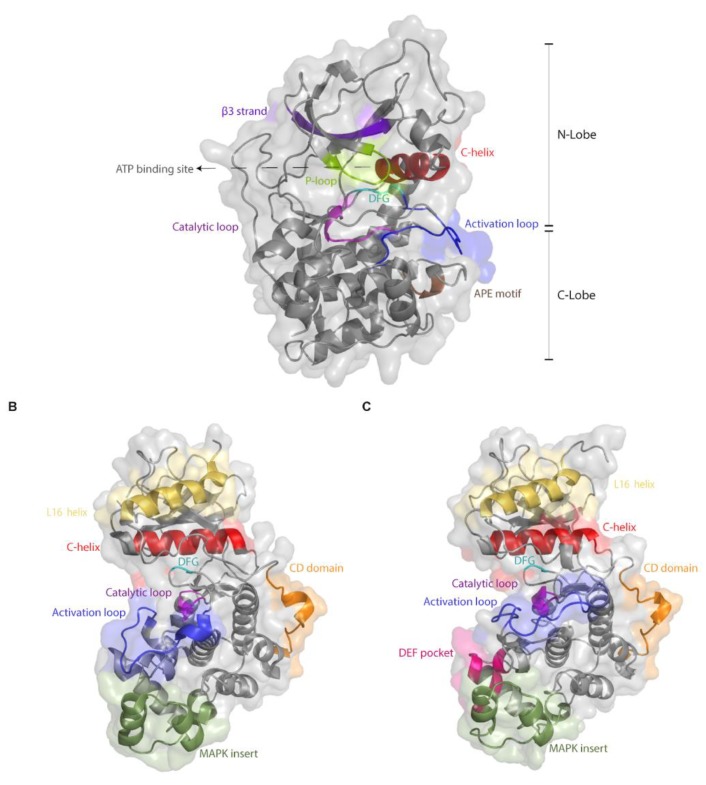
The kinase fold of Erks is highly similar to that of other ePKs, but they possess additional, specific domains. Shown are the crystal structures of (**A**) PKA (PDB 1FMO), (**B**) unphosphorylated Erk2 (PDB 4S31) and (**C**) dually phosphorylated Erk2 (PDB 2ERK). All panels show a cartoon representation covered with a transparent molecular surface with important regions presented and colored accordingly. Note the L16 helix and MAPK insert, not present in PKA, and the DEF pocket that forms only in phosphorylated Erk2.

**Figure 2 cells-09-00129-f002:**
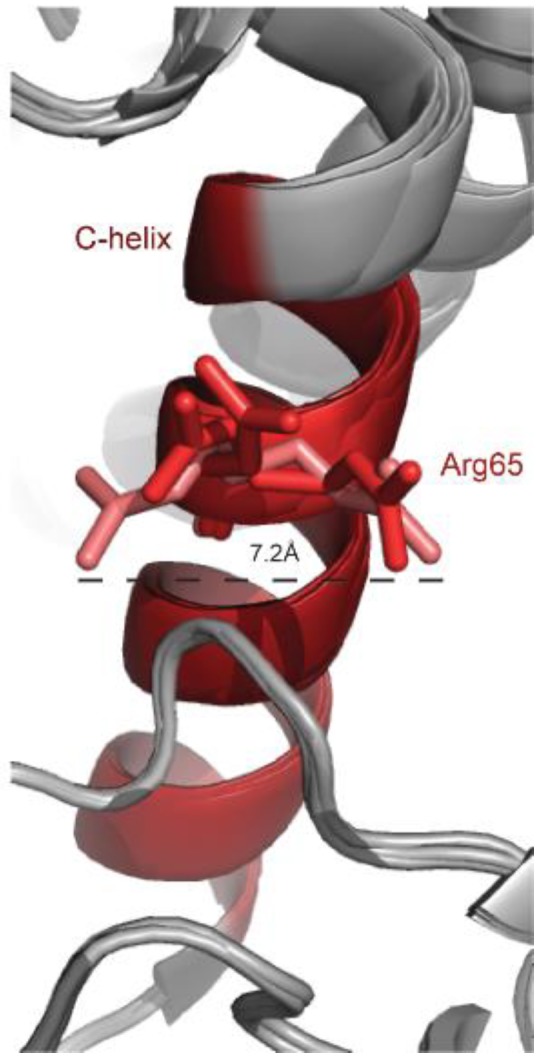
Unusual flexibility of the Arg65 residue at the αC-helix of different Erk2 crystal structures. 5 different crystal structures of Erk2 (PDB 1ERK, 4ERK, 4S31, 4GT3 and 5UMO) were superimposed and a zoom in into the αC-helix (colored in red) is presented. Arg65 is shown in sticks and the distance between the two extreme orientations is calculated.

**Table 1 cells-09-00129-t001:** (**A**) Mutations identified in ERK1 (MAPK3; mutations numeration is according to the sequence of the human ERK1). (**B**) Mutations identified in ERK2 (MAPK1) (numeration of the mutants is according to the sequence of rat ERK2).

Mutation		Mode of Identification	Reference
(**A**)			
A6_Q7insA		In cancer patients	COSMIC, cBioPortal
A6dup		In cancer patients	COSMIC
Q7R		In cancer patients	COSMIC
Q7H		In cancer patients	COSMIC, cBioPortal
G10R		In cancer patients	COSMIC
G11V		In cancer patients	COSMIC
G11_Splice		In cancer patients	COSMIC, cBioPortal
G12E		In cancer patients	COSMIC
E13K		In cancer patients	COSMIC, cBioPortal
E13*		In cancer patients	COSMIC, cBioPortal
R15G		In cancer patients	COSMIC, cBioPortal
R16I		In cancer patients	COSMIC, cBioPortal
E18Q		In cancer patients	COSMIC, cBioPortal, TumorPortal
V20V		In cancer patients	COSMIC
G23G		In cancer patients, and in a screen for mutants resistant to Erk inhibitors	COSMIC, [96]
V24F		In cancer patients	COSMIC, cBioPortal
V24S		In cancer patients	cBioPortal
V24fs*8		In cancer patients	cBioPortal
P25S		In cancer patients	COSMIC, cBioPortal
E27fs*35		In cancer patients	COSMIC, cBioPortal
E27G		In cancer patients	COSMIC
M30I		In cancer patients	COSMIC, cBioPortal
G33W		In cancer patients	cBioPortal
P35S		In cancer patients	COSMIC
D37D		In cancer patients	COSMIC, TumorPortal
Q46H		In cancer patients	cBioPortal
I48N		In a screen for mutants resistant to VRT-11E	[96]
G51S		In a screen for mutants resistant to RAF/MEK inhibitors	[96]
A52P		In cancer patients	COSMIC, cBioPortal
Y53Y		In cancer patients	COSMIC
Y53H		In a screen for mutants resistant to VRT-11E and SCH772984	[96]
Y53C		In a screen for mutants resistant to Erk inhibitors	[96]
G54S		In a screen for mutants resistant to VRT-11E and SCH772984	[96]
G54C		In cancer patients	cBioPortal
G54A		In a screen for mutants resistant to Erk inhibitors	[96]
S57G		In a screen for mutants resistant to RAF/MEK inhibitors	[96]
X57_splice		In cancer patients	cBioPortal
S58L		In cancer patients	cBioPortal, TumorPortal
Y60C		In a screen for mutants resistant to Erk inhibitors	[96]
D61E		In cancer patients	COSMIC, cBioPortal
H62_S74>R		In cancer patients	COSMIC, cBioPortal
V63M		In cancer patients	COSMIC
R64C		In cancer patients	COSMIC, cBioPortal
R64L		In cancer patients	cBioPortal
K65R		In a screen for mutants resistant to Erk and RAF/MEK inhibitors	[96]
R67C		In cancer patients	COSMIC
V68L		In cancer patients	cBioPortal
K72N		In cancer patients	COSMIC, cBioPortal
K72R		In a screen for mutants resistant to Erk inhibitors	[96]
I73S		In a screen for mutants resistant to Erk inhibitors	[96]
I73M		In Cancer patients	COSMIC, cBioPortal
S74G		In a screen for mutants resistant to VRT-11E	[96]
Insertion 74N		An activating mutation. On the basis of inferred ancestor	[75]
P75P		In Cancer patients	TumorPortal
P75L		In a screen for mutants resistant to VRT-11E and SCH772984	[96]
P75S		In a screen for mutants resistant to VRT-11E and SCH772984	[96]
E77E		In cancer patients, and in a screen for mutants resistant to Erk inhibitors	COSMIC, cBioPortal, TumorPortal, [96]
Q79H		In cancer patients	cBioPortal
Q79*		In cancer patients	COSMIC, cBioPortal, TumorPortal
Y81C		In a screen for mutants resistant to VRT-11E	[96]
C82Y		In cancer patients	cBioPortal, [96]
R84H		In cancer patients, and in a screen for mutants resistant to trametinib and dabrafenib	COSMIC, cBioPortal,[96]
R84S		In a screen for mutants resistant to RAF/MEK inhibitors, and in a screen for MEK-independent mutants	[73,96]
T85T		In cancer patients	COSMIC
L86L		In cancer patients	COSMIC
L86R		In a screen for mutants resistant to RAF/MEK inhibitors	[96]
L86P		In a screen for mutants resistant to RAF/MEK inhibitors	[96]
R87W		In cancer patients	COSMIC, cBioPortal
Q90R		In a screen for mutants resistant to trametinib and dabrafenib	[96]
R94C		In cancer patients	COSMIC, cBioPortal, TumorPortal
R96H		In cancer patients	COSMIC, cBioPortal
R96C		In cancer patients	cBioPortal
R96R		In cancer patients	COSMIC
H97H		In cancer patients	COSMIC
E98K		In cancer patients	COSMIC, cBioPortal
E98D		In cancer patients	COSMIC, cBioPortal
I101I		In cancer patients	COSMIC
G102D		In cancer patients	COSMIC, cBioPortal
I103A		On the basis of structural considerations	[76]
R104Q		In cancer patients	cBioPortal
L107L		In cancer patients	TumorPortal
R108W		In cancer patients	COSMIC, cBioPortal
A109P		In cancer patients	COSMIC
Q122M		An activating mutation. On the basis of inferred ancestor	[75]
L124L		In cancer patients	COSMIC
E126*		In cancer patients	COSMIC, cBioPortal, TumorPortal
S135N		In a screen for mutants resistant to RAF/MEK inhibitors	[96]
Q136fs*4		In cancer patients	cBioPortal
Q136*		In cancer patients	cBioPortal
Q136H		In cancer patients	cBioPortal
L138L		In cancer patients	COSMIC
H142H		In cancer patients	COSMIC
Y145*		In cancer patients	COSMIC
Y148H		In a screen for mutants resistant to trametinib and dabrafenib	[96]
Y148C		In cancer patients	COSMIC, cBioPortal
R152L		In cancer patients	TumorPortal
R152W		In cancer patients	COSMIC, cBioPortal, TumorPortal
G153S		In cancer patients	COSMIC, cBioPortal
G153G		In cancer patients	COSMIC
H158L		In cancer patients	COSMIC
S159S		In cancer patients	COSMIC, TumorPortal
A160T		In cancer patients	COSMIC, cBioPortal
A160A		In cancer patients	COSMIC
L163L		In cancer patients	COSMIC
R165L		In cancer patients	COSMIC
P169L		In cancer patients	cBioPortal
N171D		In cancer patients	COSMIC, cBioPortal
T177I		In cancer patients	COSMIC, cBioPortal
C178R		In cancer patients	COSMIC
C178Y		In cancer patients	COSMIC
C178C		In cancer patients	COSMIC
D179N		In cancer patients	COSMIC
I182N		In cancer patients	cBioPortal
I182-splice		In cancer patients	TumorPortal
F185F		In cancer patients	COSMIC
F185I		In cancer patients	cBioPortal
G186R		In cancer patients	COSMIC, cBioPortal, TumorPortal
G186D		In a screen for mutants resistant to VRT-1E and SCH772984	[96,97]
R189W		In cancer patients	COSMIC, cBioPortal
R189Q		In cancer patients	COSMIC
R189R		In cancer patients	COSMIC
I190T		In cancer patients	COSMIC, cBioPortal
P193T		In cancer patients	COSMIC, cBioPortal
P193H		In cancer patients	COSMIC, cBioPortal
P193S		In cancer patients	cBioPortal, TumorPortal
E194Q		In cancer patients	COSMIC, cBioPortal
T198T		In cancer patients	COSMIC
G199D		In cancer patients	COSMIC, cBioPortal
T202M		In cancer patients	COSMIC, cBioPortal
E203K		In cancer patients	cBioPortal
A206V		In a screen for mutants resistant to trametinib and dabrafenib	[96]
T207T		In cancer patients	COSMIC, TumorPortal
R211Q		In cancer patients	COSMIC, cBioPortal
R211P		In cancer patients	COSMIC, cBioPortal
R211W		In cancer patients	cBioPortal
E214D		In cancer patients	cBioPortal
M216I		In cancer patients, and in a screen for mutants resistant to RAF/MEK inhibitors	COSMIC, cBioPortal,[96]
N218N		In cancer patients	COSMIC
S219F		In cancer patients	COSMIC, cBioPortal
S219P		In a screen for mutants resistant to trametinib and dabrafenib	[96]
X221_splice		In cancer patients	cBioPortal
D227N		In cancer patients	cBioPortal
V231L		In cancer patients	COSMIC
A236T		In cancer patients	COSMIC
S240C		In cancer patients	COSMIC
R242R		In cancer patients	TumorPortal
L251L		In cancer patients	COSMIC
Q253P		In cancer patients	COSMIC, cBioPortal
I257V		In cancer patients	COSMIC, cBioPortal
I260H		In cancer patients	COSMIC, cBioPortal
Q266*		In cancer patients	COSMIC, cBioPortal
L269P		In cancer patients	COSMIC, cBioPortal, TumorPortal
I273M		In Cancer patients	cBioPortal
R278*		In cancer patients, and in a screen for mutants resistant to RAF/MEK inhibitors	cBioPortal, [96]
L281I		In cancer patients	cBioPortal
V290A		In cancer patients	COSMIC, cBioPortal, TumorPortal
F296F		In cancer patients	COSMIC
D300E		In cancer patients	COSMIC, cBioPortal
A303V		In cancer patients, and in a screen for mutants resistant to RAF/MEK inhibitors	COSMIC, cBioPortal, [96]
L304P		In Cancer patients	cBioPortal
L306L		In Cancer patients, and in a screen for mutants resistant to Erk inhibitors	cBioPortal, [96]
T312S		In cancer patients	cBioPortal
N314I		In cancer patients	COSMIC, cBioPortal, TumorPortal
N314N		In cancer patients	COSMIC
P315H		In cancer patients	COSMIC
R318W		In cancer patients	COSMIC, cBioPortal
R318R		In cancer patients	COSMIC
V321fs*40		In cancer patients	cBioPortal
V321W		In cancer patients	cBioPortal
P328L		In cancer patients	COSMIC, cBioPortal
E331E		In cancer patients	COSMIC
Q332H		In cancer patients	COSMIC
D335N		In cancer patients	COSMIC, cBioPortal
P336Q		In cancer patients	COSMIC
T337T		In cancer patients	COSMIC
E339V		In cancer patients	COSMIC, cBioPortal
E343K		In cancer patients	cBioPortal
P345T		In cancer patients	COSMIC
F346I		In cancer patients	COSMIC, cBioPortal
F346F		In cancer patients	COSMIC
F348F		In cancer patients	COSMIC
A349T		In cancer patients	cBioPortal
R359W		In cancer patients	COSMIC, cBioPortal, TumorPortal
R359Q		In cancer patients	COSMIC
R359L		In cancer patients	COSMIC, cBioPortal
E362K		In cancer patients	COSMIC, cBioPortal
E362*		In cancer patients	COSMIC, TumorPortal
F365C		In cancer patients	cBioPortal
Q366H		In cancer patients	cBioPortal
E367D		In cancer patients	COSMIC
P373P		In cancer patients	COSMIC, TumorPortal
G374*		In cancer patients	COSMIC
G374K		In cancer patients	COSMIC, cBioPortal
L376R		In cancer patients	COSMIC, cBioPortal
A378G		In cancer patients	COSMIC, cBioPortal
(**B**)			
A2V	In cancer patients		COSMIC, cBioPortal
A6_A7delAA	In cancer patients		COSMIC
A5delA	In cancer patients		COSMIC
A7T	In cancer patients		COSMIC, cBioPortal
G8S	In cancer patients		COSMIC, cBioPortal
R13P	In a screen for mutants that are resistant to VRT-11E and SCH772984		[95]
Q15Q	In cancer patients		COSMIC
P21S	In cancer patients		COSMIC, cBioPortal
I29M	In Cancer patients, and in a screen for mutants that are resistant to VRT-11E		COSMIC, cBioPortal,[95]
I29Q	In a screen for mutants that are resistant to VRT-11E		[95]
I29R	In a screen for mutants that are resistant to VRT-11E		[95]
I29L	In a screen for mutants that are resistant to VRT-11E		[95]
I29E	In a screen for mutants that are resistant to VRT-11E		[95]
I29K	In a screen for mutants that are resistant to VRT-11E		[95]
I29H	In a screen for mutants that are resistant to VRT-11E		[95]
I29Y	In a screen for mutants that are resistant to VRT-11E		[95]
I29D	In a screen for mutants that are resistant to VRT-11E		[95]
I29C	In a screen for mutants that are resistant to VRT-11E		[95]
I29W	In a screen for mutants that are resistant too VRT-11E		[95]
I29N	In a screen for mutants that are resistant to VRT-11E		[95]
G30P	In a screen for mutants that are resistant to VRT-11E		[95]
E31P	In a screen for mutants that are resistant to SCH772984		[95]
E31Q	In cancer patients		COSMIC, cBioPortal
G32D	In cancer patients		COSMIC, cBioPortal
G32C	In cancer patients		cBioPortal
A33N	In a screen for mutants that are resistant to VRT-11E		[95]
Y34Y	In cancer patients		COSMIC
Y34H	In a screen for mutants that are resistant to VRT-11E and SCH772984		[95,96]
Y34V	In a screen for mutants that are resistant to VRT-11E and SCH772984		[95]
Y34T	In a screen for mutants that are resistant to VRT-11E and SCH772984		[95]
Y34Q	In a screen for mutants that are resistant to VRT-11E and SCH772984		[95]
Y34G	In a screen for mutants that are resistant to VRT-11E and SCH772984		[95]
Y34S	In a screen for mutants that are resistant to VRT-11E and SCH772984		[95]
Y34C	In a screen for mutants that are resistant to VRT-11E and SCH772984		[95]
Y34I	In a screen for mutants that are resistant to VRT-11E and SCH772984		[95]
Y34D	In a screen for mutants that are resistant to VRT-11E and SCH772984		[95]
Y34R	In a screen for mutants that are resistant to VRT-11E and SCH772984		[95]
Y34N	In a screen for mutants that are resistant to VRT-11E and SCH772984		[95,96]
Y34L	In a screen for mutants that are resistant to VRT-11E and SCH772984		[95]
Y34M	In a screen for mutants that are resistant to VRT-11E and SCH772984		[95]
G35D	In a screen for mutants that are resistant to VRT-11E and SCH772984		[95]
G35T	In a screen for mutants that are resistant to VRT-11E and SCH772984		[95]
G35K	In a screen for mutants that are resistant to VRT-11E and SCH772984		[95]
G35S	In a screen for mutants that are resistant to VRT-11E and SCH772984		[95,96]
G35A	In a screen for mutants that are resistant to VRT-11E and SCH772984		[95]
G35N	In a screen for mutants that are resistant to VRT-11E		[95]
G35P	In a screen for mutants that are resistant to VRT-11E		[95]
G35C	In a screen for mutants that are resistant to VRT-11E and SCH772984		[95]
M36P	In a screen for mutants that are resistant to VRT-11E		[95]
V37A	In a screen for mutants that are resistant to VRT-11E		[95]
C38P	In a screen for mutants that are resistant to VRT-11E and SCH772984		[95]
C38R	In a screen for mutants that are resistant to SCH772984		[95]
Y41W	In a screen for mutants that are resistant to SCH772984		[95]
Y41E	In a screen for mutants that are resistant to SCH772984		[95]
D42H	In a screen for mutants that are resistant to VRT-11E and SCH772984		[95]
V44F	In cancer patients		cBioPortal
V47A	In a screen for mutants that are resistant to SCH772984		[95]
R48*	In cancer patients		COSMIC, cBioPortal
V49K	Causes resistance to SCH772984		[95]
V49H	In a screen for mutants that are resistant to VRT-11E and SCH772984		[95]
A50S	In Cancer patients		COSMIC, TumorPortal
K53G	In a screen for mutants that are resistant to SCH772984		[95]
I54H	In a screen for mutants that are resistant to SCH772984		[95]
I54D	In a screen for mutants that are resistant to SCH772984		[95]
54W	In a screen for mutants that are resistant to SCH772984		[95]
I54K	In a screen for mutants that are resistant to SCH772984		[95]
I54Y	In a screen for mutants that are resistant to SCH772984		[95]
I54E	In a screen for mutants that are resistant to SCH772984		[95]
I54Q	In a screen for mutants that are resistant to VRT-11E and SCH772984		[95]
I54S	In a screen for mutants that are resistant to VRT-11E and SCH772984		[95]
I54G	In a screen for mutants that are resistant to VRT-11E and SCH772984		[95]
I54P	In a screen for mutants that are resistant to VRT-11E		[95]
S55P	In a screen for mutants that are resistant to VRT-11E		[95]
S55G	In a screen for mutants that are resistant to VRT-11E		[95]
S55F	In a screen for mutants that are resistant to VRT-11E and SCH772984		[95]
P56L	In cancer patients, In a screen for mutants that are resistant to VRT-11E and SCH772984		COSMIC, cBioPortal, [95,96]
P56S	In a screen for mutants that are resistant to VRT-11E and SCH772984		[95,96]
P56W	In a screen for mutants that are resistant to VRT-11E and SCH772984		[95]
P56R	In a screen for mutants that are resistant to VRT-11E and SCH772984		[95]
P56K	In a screen for mutants that are resistant to VRT-11E and SCH772984		[95]
P56A	In a screen for mutants that are resistant to VRT-11E and SCH772984		[95]
P56M	In a screen for mutants that are resistant to VRT-11E and SCH772984		[95]
P56N	In a screen for mutants that are resistant to VRT-11E and SCH772984		[95]
P56G	In a screen for mutants that are resistant to VRT-11E and SCH772984		[95]
P56Y	In a screen for mutants that are resistant to VRT-11E and SCH772984		[95]
P56F	In a screen for mutants that are resistant to VRT-11E and SCH772984		[95]
P56Q	In a screen for mutants that are resistant to VRT-11E		[95]
P56V	In a screen for mutants that are resistant to VRT-11E		[95]
P56T	In a screen for mutants that are resistant to VRT-11E		[95,96]
P56I	In a screen for mutants that are resistant to VRT-11E		[95]
F57G	In a screen for mutants that are resistant to VRT-11E		[95]
F57P	In a screen for mutants that are resistant to VRT-11E		[95]
F57S	In a screen for mutants that are resistant to VRT-11E and SCH772984		[95]
F57R	In a screen for mutants that are resistant to VRT-11E and SCH772984		[95]
E58Q	In a screen for mutants that are resistant to SCH772984		[95]
E58S	In a screen for mutants that are resistant to SCH772984		[95]
Q60P	In a screen for mutants that are resistant to VRT-11E and SCH772984		[95]
T61T	In cancer patients		COSMIC
T61I	In cancer patients		COSMIC
Y62G	In a screen for mutants that are resistant to VRT-11E		[95]
Y62E	In a screen for mutants that are resistant to VRT-11E		[95]
Y62D	In a screen for mutants that are resistant to VRT-11E		[95]
Y62S	In a screen for mutants that are resistant to VRT-11E		[95]
Y62C	In a screen for mutants that are resistant to VRT-11E		[95]
Y62T	In a screen for mutants that are resistant to VRT-11E		[95]
Y62Q	In a screen for mutants that are resistant to VRT-11E		[95]
Y62A	In a screen for mutants that are resistant to VRT-11E		[95]
Y62P	In a screen for mutants that are resistant to VRT-11E		[95]
Y62V	In a screen for mutants that are resistant to VRT-11E and SCH772984		[95]
Y62M	In a screen for mutants that are resistant to VRT-11E and SCH772984		[95]
Y62K	In a screen for mutants that are resistant to VRT-11E and SCH772984		[95]
Y62I	In a screen for mutants that are resistance to VRT-11E and SCH772984		[95]
Y62L	In a screen for mutants that are resistance to VRT-11E and SCH772984		[95]
Y62R	In a screen for mutants that are resistant to VRT-11E and SCH772984		[95]
Y62N	In a screen for mutants that are resistant to VRT-11E		[95,96]
C63F	In a screen for mutants that are resistant to VRT-11E		[95]
C63W	In a screen for mutants that are resistant to VRT-11E		[95]
C63fs*3	In Cancer patients		COSMIC, cBioPortal
C63Y	In a screen for mutants that are resistant to VRT-11E and SCH772984		[95,96]
R65I	In cancer patients, and in a screen for mutants that are resistant to VRT-11E		COSMIC, cBioPortal,[95]
R65K	In a screen for mutants that are resistant to Erk inhibitors and RAF/MEK inhibitors		[96]
R65S	Genetic screen for Mpk1 intrinsically active mutants		[73]
T66T	In Cancer patients		COSMIC
T66M	In a screen for mutants that are resistant to VRT-11E		[95,96]
T66Q	In a screen for mutants that are resistant to VRT-11E		[95,96]
T66F	In a screen for mutants that are resistant to VRT-11E		[95,96]
T66I	In a screen for mutants that are resistant to VRT-11E		[95,96]
T66L	In a screen for mutants that are resistant to VRT-11E		[95,96]
T66P	In a screen for mutants that are resistant to VRT-11E		[95,96]
T66D	In a screen for mutants that are resistant to VRT-11E and SCH772984		[95,96]
T66Y	In a screen for mutants that are resistant to VRT-11E and SCH772984		[95,96]
T66H	In a screen for mutants that are resistant to VRT-11E		[95,96]
T66N	In a screen for mutants that are resistant to VRT-11E and SCH772984		[95,96]
L67L	In cancer patients		COSMIC
R68R	In cancer patients		COSMIC
E69P	In a screen for mutants that are resistant to VRT-11E		[95,96]
E69C	In a screen for mutants that are resistant to VRT-11E		[95,96]
E69G	In a screen for mutants that are resistant to VRT-11E		[95,96]
E69K	In a screen for mutants that are resistant to VRT-11E and SCH772984		[95,96]
E69A	In a screen for mutants that are resistant to VRT-11E and SCH772984		[95,96]
I72fs*8	In cancer patients		COSMIC
L73E	In a screen for mutants that are resistant to VRT-11E		[95,96]
L73H	In a screen for mutants that are resistant to VRT-11E		[95,96]
L73R	In a screen for mutants that are resistant to VRT-11E		[95,96]
L73W	In a screen for mutants that are resistant to VRT-11E and SCH772984		[95]
L73P	In a screen for mutants that are resistant to VRT-11E, and on the basis of genetic screen for intrinsically active FUS3		[95,103]
R75C	In cancer patients		COSMIC, cBioPortal
R77S	In cancer patients		cBioPortal
R77K	In cancer patients		COSMIC, cBioPortal
E79K	In cancer patients		COSMIC, cBioPortal, TumorPortal
N80fs*18	In cancer patients		COSMIC, cBioPortal
I82T	In cancer patients		COSMIC, cBioPortal
I82A	Activating mutation. On the basis of structural considerations		[76]
I84A	Activating mutation. On the basis of structural considerations		[76]
D86-del	In cancer patients		cBioPortal
I88F	In cancer patients		COSMIC
I93I	In cancer patients		COSMIC, TumorPortal
Q95R	In a screen for mutants that are resistant to VRT-11E		[95]
M96W	In a screen for mutants that are resistant to VRT-11E		[95]
M96I	In cancer patients		cBioPortal
D98N	In cancer patients		cBioPortal
D98M	In a screen for mutants that are resistant to VRT-11E		[95]
I101Q	In a screen for mutants that are resistant to SCH772984		[95]
I101W	In a screen for mutants that are resistant to SCH772984		[95]
I101Y	In a screen for mutants that are resistant to SCH772984		[95]
I101R	In cancer patients		COSMIC
V102V	In a screen for mutants that are resistant to Erk inhibitors and RAF/MEK inhibitors		[96]
Q103A	Generated in order to study the biological effect the gatekeeper residue		[76]
Q103G	Generated in order to study the biological effect the gatekeeper residue		[76]
Q103I	In a screen for mutants that are resistant to SCH772984		[95]
Q103F	In a screen for mutants that are resistant to SCH772984		[95]
Q103T	In a screen for mutants that are resistant to SCH772984		[95]
Q103W	In a screen for mutants that are resistant to VRT-11E and SCH772984		[95]
Q103V	In a screen for mutants that are resistant to VRT-11E and SCH772984		[95]
Q103N	In a screen for mutants that are resistant to VRT-11E		[95]
Q103Y	In a screen for mutants that are resistant to VRT-11E and SCH772984		[95]
D104D	In a screen for mutants that are resistant to Erk inhibitors		[96]
D104H	In cancer patients		cBioPortal
T108P	In a screen for mutants that are resistant to VRT-11E		[95]
L110R	In cancer patients		cBioPortal
L114S	In cancer patients		cBioPortal
L119I	In cancer patients		COSMIC, cBioPortal, TumorPortal
D122T	In a screen for mutants that are resistant to VRT-11E and SCH772984		[95]
I124F	In cancer patients		COSMIC
C125I	In a screen for mutants that are resistant to VRT-11E and SCH772984		[95]
Y129N	In a screen for mutants that are resistant to RAF/MEK inhibitors		[96]
Y129H	In a screen for mutants that are resistant to RAF/MEK inhibitors		[96]
Y129F	In a screen for mutants that are resistant to trametinib and dabrafenib		[96]
Y129C	In Cancer patients, and In a screen for mutants that are resistant to trametinib and dabrafenib		COSMIC, cBioPortal, [96]
Y129S	In a screen for mutants that are resistant totrametinib and dabrafenib		[96]
Q130E	In cancer patients		COSMIC, cBioPortal
L132P	In cancer patients		COSMIC, cBioPortal
R133K	In cancer patients		COSMIC, cBioPortal
G134E	In cancer patients		COSMIC, cBioPortal
I138I	In cancer patients		COSMIC
H139Y	In cancer patients		COSMIC, cBioPortal, TumorPortal
H139R	In cancer patients		cBioPortal
S140L	In cancer patients		COSMIC, cBioPortal
A141A	In cancer patients		COSMIC, TumorPortal
N142K	In cancer patients		COSMIC, cBioPortal
N142N	In cancer patients		COSMIC
H145Y	In cancer patients		COSMIC, cBioPortal
H145R	In cancer patients		cBioPortal
R146S	In cancer patients		COSMIC, cBioPortal
R146L	In cancer patients		cBioPortal, TumorPortal
R146C	In cancer patients		cBioPortal
R146H	In cancer patients		COSMIC, cBioPortal
D147Y	In cancer patients		cBioPortal
S151D	On the basis of alignment with MKK1		[103]
L154N	In a screen for mutants that are resistant to VRT-11E		[95]
L154G	In a screen for mutants that are resistant to VRT-11E		[95]
L155L	In cancer patients		COSMIC, TumorPortal
D160N	In cancer patients		COSMIC, cBioPortal, TumorPortal
D160G	In cancer patients		COSMIC, cBioPortal, TumorPortal
I163I	In cancer patients		cBioPortal
C164R	In cancer patients		COSMIC, cBioPortal
D165G	In cancer patients		COSMIC, cBioPortal
G167D	In a screen for mutants that are resistant to VRT-11E and SCH772984		[95,97]
L168L	In a screen for mutants that are resistant to Erk inhibitors		[96]
R170H	In cancer patients		COSMIC, cBioPortal, TumorPortal
V171I	In cancer patients		COSMIC
P174T	In cancer patients		COSMIC, cBioPortal
P174S	In cancer patients		COSMIC, cBioPortal
D177G	In cancer patients		cBioPortal
T179fs*29	In cancer patients		COSMIC, cBioPortal
F181S	In cancer patients		COSMIC
E184*	In cancer patients		cBioPortal
A187V	In a screen for mutants that are resistant to trametinib and dabrafenib		[96]
R189C	In cancer patients		COSMIC, cBioPortal
R189H	In cancer patients		COSMIC, cBioPortal
W190L	In cancer patients		cBioPortal
E195*	In cancer patients		cBioPortal
L198F	In cancer patients		COSMIC
S200P	In a screen for mutants that are resistant to trametinib and dabrafenib		[96]
G202C	In cancer patients		COSMIC, cBioPortal,[96]
G202S	In cancer patients		cBioPortal,[96]
G202G	In cancer patients		COSMIC, [96]
Y203N	In a screen for mutants that are resistant to RAF/MEK inhibitors		[96]
T204I	In cancer patients		COSMIC, cBioPortal
I209V	In cancer patients		COSMIC, cBioPortal
V212A	In a screen for mutants that are resistant to RAF/MEK inhibitors		[96]
E218K	In cancer patients		COSMIC
S220Y	In cancer patients		COSMIC, cBioPortal
I225F	In cancer patients		COSMIC, cBioPortal
P227S	In cancer patients		COSMIC
P227L	In cancer patients		COSMIC, cBioPortal
G228E	In cancer patients		COSMIC, cBioPortal
D233E	In cancer patients		COSMIC, cBioPortal
D233G	In cancer patients		cBioPortal
D233V	In cancer patients		COSMIC, cBioPortal
D233*	In cancer patients		TumorPortal
G240_splice	In cancer patients		cBioPortal
L242I	In cancer patients		COSMIC, cBioPortal
L242F	In cancer patients		COSMIC, cBioPortal, TumorPortal
S244F	In cancer patients		COSMIC, cBioPortal
S244S	In cancer patients		COSMIC
L250L	In a screen for mutants that are resistant to RAF/MEK inhibitors		[96]
N255S	In cancer patients		COSMIC, cBioPortal
R259G	In cancer patients		COSMIC, cBioPortal
N260T	In cancer patients		COSMIC, cBioPortal, TumorPortal
Y261C	In cancer patients		COSMIC, cBioPortal
L263F	In a screen for mutants that are resistant to Erk inhibitors		[96]
S264F	In cancer patients		COSMIC
L265P	In cancer patients		COSMIC
P266L	In cancer patients		COSMIC, cBioPortal
P272S	In cancer patients		COSMIC
L276L	In cancer patients		COSMIC, TumorPortal
L276M	In cancer patients		cBioPortal
F277F	In cancer patients		COSMIC
K283T	In cancer patients		cBioPortal
L285M	In a screen for mutants that are resistant to RAF/MEK inhibitors		[96]
L288L	In cancer patients		COSMIC
D289G	In cancer patients		COSMIC, cBioPortal, TumorPortal
D289H	In cancer patients		cBioPortal
P296T	In cancer patients		COSMIC
E301K	In a screen for mutants that are resistant to RAF/MEK inhibitors		[96]
A305S	In cancer patients		cBioPortal
L311P	In cancer patients		COSMIC
Y314F	In cancer patients		COSMIC, cBioPortal, TumorPortal
Y314C	In cancer patients		cBioPortal
D316N	In cancer patients		COSMIC, cBioPortal
P317S	In cancer patients		COSMIC, cBioPortal
P317P	In cancer patients		COSMIC
S318C	In cancer patients		COSMIC
D319N	In cancer patients, and in a genetic screen in *Drosophila*		COSMIC, cBioPortal, [77]
D319A	In cancer patients		COSMIC
D319G	In a screen for mutants that are resistant to trametinib and dabrafenib		cBioPortal, [96]
D319V	In cancer patients		COSMIC, cBioPortal
D319E	In cancer patients		COSMIC, cBioPortal
E320K	In a screen for mutants that are resistant to trametinib and dabrafenib		CISMIC, cBioPortal, TumorPortal,[96]
E320*	In cancer patients		cBioPortal, TumorPortal
E320N	In cancer patients		cBioPortal
E320A	In cancer patients		COSMIC, cBioPortal
E320V	In cancer patients		COSMIC
P321P	In a screen for mutants that are resistant to Erk inhibitors		[96]
P321S	In a screen for mutants that are resistant to RAF/MEK inhibitors		[96]
P321L	In a screen for mutants that are resistant to RAF/MEK inhibitors		[96]
I322V	In cancer patients		cBioPortal
A323T	In cancer patients		cBioPortal
A323S	In cancer patients		cBioPortal
A323V	In cancer patients		cBioPortal
E324*	In cancer patients		COSMIC, cBioPortal
F329F	In cancer patients		COSMIC
F329Y	In cancer patients		COSMIC
D330N	In cancer patients		COSMIC, cBioPortal
M331I	In cancer patients		COSMIC, cBioPortal
D335N	In a screen for mutants that are resistant to RAF/MEK inhibitors		[96]
E343*	In cancer patients		COSMIC
I345H	In a screen for mutants that are resistant to to VRT-11E		[95]
I345M	In a screen for mutants that are resistant to VRT-11E		[95]
I345F	In a screen for mutants that are resistant to VRT-11E		[95]
I345L	In a screen for mutants that are resistant to VRT-11E		COSMIC
I345Y	In a screen for mutants that are resistant to VRT-11E and SCH772984		[95]
I345W	In a screen for mutants that are resistant to VRT-11E and SCH772984		[95]
E347*	In cancer patients		COSMIC
E347K	In cancer patients, and in a screen for mutants that are resistant to RAF/MEK inhibitors		COSMIC, cBioPortal,[96]
T349T	In cancer patients		COSMIC
A350S	In cancer patients		COSMIC
A350V	In cancer patients		COSMIC
R351K	In a screen for mutants that are resistant to RAF/MEK inhibitors		[96]
Y356D	In cancer patients		COSMIC, cBioPortal
Y356Y	In cancer patients		COSMIC
R357T	In cancer patients		cBioPortal

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
