# Peer review of "Mutations That Confer Drug-Resistance, Oncogenicity and Intrinsic Activity on the ERK MAP Kinases—Current State of the Art"

_cells, 2020, doi:10.3390/cells9010129_

Round 1
Reviewer 1 Report
This is an extensively thorough, proficient, educational and entertaining piece of work on the current knowledge on ERK mutations and its biochemical/ biological implications, offering deep structure-function insights. Surely it will be most welcomed by the community as it tackles an ERK aspect hardly covered in previous reviews.
My only criticism is that I miss a short "conclusions" section, to sum top all that is told. As is, the essay ends rather abruptly.
Author Response
We thank the reviewer for the comment.
As suggested by the reviewer a final paragraph entitled 'Conclusions' was added.
Reviewer 2 Report
This is a very thorough review on ERK1/ERK2, covering topics that include the structure of the ERKs, mutations that are found in these ERKs, identified either in laboratory screening or in cancer patients, and the issue of what these mutations do to intrinsic activity of the ERKs and to cell function e.g. proliferation and to sensitivity to ERK inhibitors. The structure of the review is very well-planned, logical and the text is easy to follow. Perhaps, it would be worthwhile commenting on the types of cancer in which the patient mutations in ERKs were found. There are some minor issues that the authors can easily address.
An old reference (18) is provided in the review. Perhaps, a recent review covering the topic is more appropriate in this instance. Line 456: It would be good to mention that VRT-11E is an ERK inhibitor at line 456, instead of mentioning it at line 461. Line 488: Please add "SCH772984, another ERK inhibitor" Line 374: suppression should be suppressing Line 509: resistance instead of resistant Line 543: missing "they"Author Response
We thank the reviewer for the comment.
We have accepted all changes suggested by the reviewer:
Ref. 18 was replaced. The suggested wording changes in lines 456, 461, 488, 374, 509 and 543 were made exactly as the reviewer proposed. The types of cancers in which the mutations were discovered are mentioned in lines 221 and 442.